# Multi-Dimensional Elimination of β-Lactams in the Rural Wetland: Molecule Design and Screening for More Antibacterial and Degradable Substitutes

**DOI:** 10.3390/molecules27238434

**Published:** 2022-12-02

**Authors:** Shuhai Sun, Zhuang Li, Zhixing Ren, Yu Li

**Affiliations:** 1School of Hydraulic and Environmental Engineering, Changchun Institute of Technology, Changchun 130012, China; 2College of Forestry, Northeast Forestry University, Harbin 150040, China; 3MOE Key Laboratory of Resources and Environmental Systems Optimization, North China Electric Power University, Beijing 102206, China

**Keywords:** β-lactams, combined biodegradation, molecular docking, molecular modification, constructed wetland, molecular dynamics simulation

## Abstract

Restricted economic conditions and limited sewage treatment facilities in rural areas lead to the discharge of small-scale breeding wastewater containing higher values of residual beta-lactam antibiotics (β-lactams), which seriously threatens the aquatic environment. In this paper, molecular docking and a comprehensive method were performed to quantify and fit the source modification for the combined biodegradation of β-lactams. Using penicillin (PNC) as the target molecule, combined with contour maps for substitute modification, a three-dimensional quantitative structure–activity relationship (3D-QSAR) model was constructed for the high-performance combined biodegradation of β-lactams. The selected candidate with better environmental friendliness, functionality, and high performance was screened. By using the homology modeling algorithms, the mutant penicillin-binding proteins (PBPs) of *Escherichia coli* were constructed to have antibacterial resistance against β-lactams. The molecular docking was applied to obtain the target substitute by analyzing the degree of antibacterial resistance of β-lactam substitute. The combined biodegradation of β-lactams and substitute in the constructed wetland (CW) by different wetland plant root secretions was studied using molecular dynamics simulations. The result showed a 49.28% higher biodegradation of the substitutes than PNC when the combined wetland plant species of *Eichhornia crassipes*, *Phragmites australis*, and *Canna indica L*. were employed.

## 1. Introduction

The aquatic environment, a typical site for antibiotic enrichment, is central to the environmental behavior of antibiotic production [1]. β-lactams, one of the widely used feed additives in the breeding industry, is a commonly used antibiotic [2]. Animals excrete β-lactams as original compounds because of intestinal malabsorption or incomplete metabolism [3]. Thus, the residual β-lactams are released as “pseudo persistent” pollution in the aquatic environment through the breeding wastewater [4], which also causes serious non-point source pollution [5]. Since the population in rural China is relatively dispersed, which is accompanied by limited economic conditions, sewage collection, and poor infrastructure, the discharge of small-scale breeding facilities is not well treated [5].

In natural environments, the degradation of β-lactams happens through hydrolysis, photodegradation, adsorption, and biodegradation, among which biodegradation is an important pathway [4]. Both plants and microbes assist in biodegradation, in which microbial degradation by itself is unideal due to environmental conditions [6]. In addition to microbial degradation, plant absorptivity, mineralization, and root secretions also affect the biodegradation efficiency of antibiotic molecules [7]. Gu et al. used molecularly directed modification to identify key amino acid residues of proteins that affect the uptake, degradation, and mineralization of polychlorinated naphthalene in plants. Then, by using various molecular biological methods, mutation sites were precisely designed to alter specific amino acid residues in the target proteins, improving the properties of protein molecules [7]. Chen et al. degraded oxytetracycline (OTC) using a combined plant-microbial system consisting of *Phyllobacterium myrsinacearum*, *Rhodotorula mucilaginosa*, *Tagetes patula* L., and *Mirabilis jalapa* L., in which OTC degradation rates of 31–71% were achieved when the OTC mass fraction was 5 mg/kg [6]. Zhou et al. constructed a novel water treatment scheme by combining *Microcystis aeruginosa* with plants to degrade four antibiotics in urban rivers with 70–95% of total antibiotics degraded [8]. Al-Ahmad confirmed that aquatic plants enhanced sulfonamides degradation and removal in the aquatic environment (91–99%, 73–92%, and 95–99% for sulfadiazine, sulfamethazine, and sulfamethoxazole, respectively) [9]. The synergistic degradation of β-lactams using a combined plant–microbial system has the advantages of low carbon, higher degradation efficiency, and is more economical [10]. Therefore, the use of combined β-lactams degradation by plant–microorganism has become one of the mainstream ideas.

For the removal of antibiotics from breeding wastewater, the physical adsorption method has the advantages of simple equipment, a fast and efficient process, and convenient operation and management; however, it is difficult for activated carbon to selectively adsorb antibiotics from breeding wastewater with a high concentration of organic pollutants [11]. Although the chemical oxidation method has the advantages of high antibiotic clearance rate and strong selectivity in the treatment of antibiotics in wastewater, the added chemical oxidants can easily cause secondary pollution to the aquatic environment [12,13]. Activated sludge technology mainly uses the metabolic action of microorganisms to decompose antibiotics in sewage. Although activated sludge technology has mild reaction conditions and strong operability, the growth environment of microorganisms needs to be strictly controlled, the equipment cost is high, and the treatment range is narrow [14]. The hydrolysis of antibiotics in sewage is not only limited by the temperature and pH of the aquatic environment, but also the influence of ionic strength on the hydrolysis rate of antibiotics is complex and changeable. Different ionic strengths may promote or inhibit the hydrolysis of antibiotic molecules, resulting in the difficulty for humans to control the hydrolysis reaction of antibiotics in sewage [15,16]. The photolysis of antibiotics in sewage is limited by the light absorption capacity of antibiotics and the types of functional groups, etc. For example, penicillin in β-lactams has a weak photolysis reaction in water due to its weak absorption intensity in the near ultraviolet and visible areas [17]. Compared with traditional treatments and natural degradation, constructed wetland has an efficient and stable treatment effect on the removal of antibiotics from breeding wastewater, and on this basis, it has sustainable ecology, better environmental friendliness and lower treatment cost [18]. The efficient removal of antibiotics from breeding wastewater by CW is enabled by the multi-path pollutant degradation and transformation characteristics of the process, which remove antibiotic components through plant absorption, filler adsorption, and microbial degradation [19]. Plants play an important role in the degradation and migration of compounds [20]. Huang et al. used CW to treat residual tetracyclic antibiotics in breeding wastewater, in which 69–99% of tetracycline was removed [21]. Choi et al. achieved an 80% removal efficiency of residual sulfathiazole in breeding wastewater by using CW [22]. Thus, the above studies represented CW as a feasible solution for degrading antibiotics.

In this work, an ecological and sustainable scheme was constructed to eliminate the potential environmental risks of β-lactams in small-scale breeding wastewater in rural areas. The main research objectives of this work involve (1) source modification dimension: design of β-lactam substitutes with better bactericidal, environmentally friendly, and biodegradable properties; (2) antibacterial resistant dimension: virtual design of resistant mutations for screening the antibacterial resistant properties of β-lactam substitute, followed by an analysis of their resistant generation and antibacterial resistant mechanisms of β-lactam substitute; (3) terminal treatment dimension: screening of selected wetland plants in the CW system to eliminate the environmental impacts of β-lactams by biodegradation.

## 2. Results and Discussion

### 2.1. Acquisition of the Comprehensive Evaluation Value of Combined Biodegradability of β-Lactams Based on CM

In this study, we used the LibDock module of the Discovery Studio 2020 software, the LibDock Score of β-lactams and receptor proteins were obtained using the molecular docking method. Then, the LibDock scores for the degradation of four antibiotics were fitted to the comprehensive evaluation value using the CM (Appendix A).

### 2.2. Molecular Modification and Evaluation of High-Performance Combined Biodegradation of β-Lactams Based on the 3D-QSAR Model

#### 2.2.1. 3D-QSAR Model Evaluation of High-Performance Combined Biodegradability of β-Lactams

In this study, the CoMSIA model for high-performance combined biodegradation of β-lactams constructed by coupling the molecular docking method, CM, and the 3D-QSAR model showed a *q*^2^ of 0.629 (>0.5), and the best principal component score *N* of 10, indicating that the model had a good predictive ability. The *R*^2^ of 1.000, SEE of 0.016, and *F* of 565,064.468 indicated that this model had a credible predictive ability and good stability [23]. The test set cross-validate coefficient *r*^2^_pred_ of 0.922 (>0.6) and standard error *SEP* of 2.229 indicated that the constructed CoMSIA model could accurately predict β-lactam substitutes with high-performance combined biodegradation [24].

#### 2.2.2. Molecular Modification of High-Performance Combined Biodegradation β-Lactam Substitutes

A total of 201 β-lactam substitutes were obtained after molecular modification (Appendix A). The high-performance combined biodegradation of the modified β-lactam substitutes was significantly more compared to the PNC (Appendix A). Among them, 18 substitutes were designed for single-site modification, and the combined biodegradation improved by 14.69% on average compared to that of the original β-lactams. A total of 53 substitutes were designed for two-site modification, and the combined biodegradation was improved by 13.34% on average compared to that of the original β-lactams. Finally, 130 substitutes were designed for three-site modification, and the combined biodegradation was improved by 16.85% on average compared to that of the original β-lactams.

#### 2.2.3. Evaluation of Environmental Friendliness of β-Lactam Substitutes and Validation of the 3D-QSAR Model

In this study, the bioconcentration, biotoxicity, and long-range transport of 201 β-lactam substitutes obtained by molecular modification were predicted and evaluated by the Estimation Programs Interface (EPI) Suite database. First, the SMILES conversion function of the NovoPro online tool was used to convert the mol format files of β-lactam and substitutes into molecular SMILES numbers. The molecular SMILES numbers of β-lactam substitutes were entered into the EPI software for predicting the biotoxicity, bioconcentration, and long-range mobility, and their predicted values were obtained.

From the EPI Suite data, the bioconcentration Log*K*_ow_ value for PNC was 1.83, the LC_50_ value for LC_50_ was 1282.44 (mg/L), and the Log*K*_oa_ value for long-range mobility was 14.15. Compounds with Log*K*_ow_ values below 1 or above 4 are not easily absorbed by plants [25]. In this study, we screened β-lactam substitutes that were weaker than the target molecules for biotoxicity and bioconcentration intensity and had some plant absorption ability. Lee et al. showed that the long-range mobility class of the compound Log*K*_oa_ > 10 is at a low migration level [26]. Therefore, we used Log*K*_ow_ prediction values between 1–1.83, LC_50_ prediction values greater than 1282.44 (mg/L), and Log*K*_oa_ greater than 10 as screening conditions for screening β-lactam substitutes to meet the evaluation criteria of environmental friendliness, and finally, screened 25 β-lactam substitutes (Appendix A).

In this study, β-lactam substitutes were docked with the receptor proteins by the molecular docking method based on the LibDock Score, to characterize their sterilization, plant absorption, plant mineralization, and microbial degradation. Based on the evaluation of environmental friendliness, 15 β-lactam substitutes with better overall properties were further screened (Table 1).

### 2.3. Screening of Antibacterial Resistance of β-Lactam Substitutes

#### 2.3.1. Construction of Drug Resistant Mutations of *E. coli* PBP

In this study, we obtained the original amino acid sequence of PBP from the PDB database. Based on the random amino acid sequence mutation module of the Sequence Manipulation Tool for specific points, we performed random amino acid mutation of the key residue “SXXK”. The amino acid sequences adjacent to it in the original amino acid sequence of the PBP (72–78, 209–214) were subjected to random amino acid mutations to obtain the amino acid sequence of the receptor protein resistant mutations [27]. The key residues of PBP resistant “SXXK” and its adjacent amino acid sequences are shown in Figure 1.

As shown in Table 2, several PBP resistant mutations were constructed using a homology modeling algorithm in this study, and the resistant mutations underwent molecular docking with PNC to characterize the degree of resistance of the mutant proteins. Finally, the most resistant PBP mutations were screened.

As shown in Table 2, the drug resistance of the mutant PBP improved significantly by mutating the key residue “SXXK” and the adjacent amino acid sequences from “ASLTKMM, YSIYKE” to “HVNWSPS, GANEVG”. The drug resistance of the mutant PBP constructed after the mutation of “SXXK” and the adjacent amino acid sequences from “ASLTKMM, YSIYKE” to “HVNWSPS, GANEVG” increased significantly, and the drug resistance of the mutant protein was 19.86% higher than that of the original PNC. Accordingly, the PBP-14 resistant mutation was selected as the bacterial drug resistant characteristic receptor for further molecular screening of the substitute.

#### 2.3.2. Evaluation and Screening of Antibacterial Resistant Properties of β-Lactam Substitutes

Fifteen β-lactam substitutes screened by environmental friendliness evaluation and comprehensive property evaluation underwent molecular docking with the PBP of the resistant mutation to evaluate the antibacterial resistance of β-lactam substitutes (Table 3).

As shown in Table 3, PNC-196 had the most significant (25.70%) improvement in the bactericidal effect, and the combined biodegradation, plant absorption, and plant mineralization of PNC-196 were satisfactory. Therefore, PNC and PNC-196 were selected as the control ligand molecules for the CW-simulated degradation of β-lactams in this study.

#### 2.3.3. Expression of Drug Resistance in *E. coli* and Analysis of the Mechanism of Resistance to β-Lactam Substitute

We selected PNC and PNC-196 with a significantly enhanced ability against resistant *E. coli* as the representatives and performed molecular docking and mechanistic analysis on PBP and its resistant mutation. The results of molecular docking showed that β-lactams and substitutes play a major role in binding to the receptor protein by amino acid residues around the receptor protein, which are bound to the receptor protein by hydrogen bonds, charge, or polar interactions (Figure 2).

As shown in Table 2, the drug resistance of PBP increased when the key residue “SXXK” and the hydrophobic non-polar amino acids in the adjacent amino acid sequences were mutated to hydrophilic polar positively charged amino acids or when the hydrophilic polar amino acids were mutated to hydrophobic non-polar amino acids, such as when “ASLTKMM, YSIYKE” was mutated to “HVNWSPS, GANEVG”. When ASLTKMM, YSIYKE” was mutated to “HVNWSPS, GANEVG”, the resistance of PBP increased by 19.86%. As shown in Table 4, when the amino acid residues of the binding process of β-lactams to PBP changed from hydrophobic amino acids to hydrophilic amino acids, the resistance of *E. coli* increased. This indicated that the binding of β-lactams to hydrophilic and hydrophobic amino acid residues also affects the resistance of PBP.

To further investigate the mechanistic role of PNC-196 against *E. coli’s* resistance, we determined the bonding characteristics between the surrounding amino acid residues with PNC-196 and its resistant mutation of PBP (Table 5). Among them, Chu et al. [28] suggested that the average distance between amino acid residues and molecules in molecular docking would directly influence the docking effect; the smaller the average distance, the better the docking effect, and vice versa. As shown in Table 5, in the docking system of β-lactams and substitute with PBP, four bonds were present between β-lactams and PBP, and six bonds were present between PNC-196 and PBP. PNC-196 was 5.01% more stable than the original drug, in resistant *E. coli*. In the docking system of PNC-196 with the PBP of resistant mutation, the ability of PNC-196 against resistant *E. coli* was significantly higher (25.70%). Based on the results of the analysis of the action bond situation between PNC-196 and the amino acid residues of PBP of the resistant mutation, we found that compared to the molecular docking between β-lactams and the proteins after mutation, the shortest action bond between PNC-196 and the proteins decreased by 36.00%, the longest action bond increased by 7.02%, and the average bond length increased by 0.59%.

### 2.4. Construction, Screening, and Characterization of CW Based on Molecular Dynamics Simulation to Eliminate the Effects of the β-Lactam System

In this study, the plant root exudates of five CW plants (Water hyacinth, Reed, Typha, Calamus, and Canna) were selected as external condition molecules and used for constructing a molecular dynamics simulation-based system for eliminating the effects of β-lactams [29], in which the root exudates of Water hyacinth included benzoic acid, propionic acid, succinic acid, malic acid, acetic acid, and tartaric acid. The root exudates of Reed included malic acid, glucose, xylose, sucrose, and tartaric acid. The root exudates of Typha included glucose, sucrose, tartaric acid, malic acid, malonic acid, and fructose. The root exudates of the Calamus included glucose, sucrose, galactose, xylose, tartaric acid, malic acid, and propionic acid. The root exudates of Canna included malic acid, tartaric acid, succinic acid, and acetic acid [29,30].

A five-factor, two-level (0 for not added, 1 for added) control variable was used for conducting an experiment with a full factorial design using plant root exudates of five wetland plants as the external condition. As Pseudomonas was the rhizosphere microbe microorganism of the Water hyacinth, we considered the Water hyacinth as the essential wetland plant in this study. In total, 32 sets of combinations of external condition molecules for the combined biodegradation of β-lactams (including the blank group) were designed and generated. Additionally, PNC and PNC-196 were used as controls to simulate the binding of PNC and PNC-196 to Pseudomonas β-lactamase. In the simulation, different combinations of external condition molecules of wetland plant root exudates were added, and the effects of different CW planting schemes on the β-lactams before and after the modification were evaluated (Figure 3).

The results of the molecular dynamics simulations showed that the binding energy of the blank control combination of β-lactams was −5.35 kJ/mol and that of the blank control combination of β-lactam substitute was −33.60 kJ/mol, PNC-196 was more degradable by microbes. The binding energy of each simulated experimental group decreased by different degrees relative to the blank control group, i.e., the binding ability of β-lactams and substitute to *Pseudomonas* β-lactamase increased, indicating that the wetland plant root secretions in each planting scheme of the CW promoted the microbial degradation of β-lactams to different degrees. Among them, the simulated experimental combination 8 of the PNC-196 planting scheme combination (Water hyacinth, Reed, and Canna) promoted microbial degradation of PNC-196 most effectively with a binding energy of −66.25 kJ/mol (49.28%). Additionally, the mean binding energy for the combination of the implantation scheme of β-lactams was −43.90 kJ/mol, and the mean binding energy for the combination of molecular implantation scheme of PNC-196 was −51.18 kJ/mol, with an average improvement of 16.58% in the degradation of β-lactams. These findings indicated that the substitute had better biodegradability than β-lactams; the average binding energy of the CW degradation of β-lactams scheme involving four wetland plants, Reed, Typha, Calamus, and Canna, at the single-factor level is shown in Figure 4. Canna contributed the most to the degradation of β-lactams in the CW, and showed the highest increase in the degradation capacity of the substitute (19.29%) (Figure 4). Reed and Canna had strong effects on the antibiotic degradation in CW. Yi et al. used Reed and Canna as test plants to construct the CW system and investigated the removal effect of tetracycline antibiotics and found that the removal of tetracycline, oxytetracycline, and chlorotetracycline in Canna and Reed wetlands under certain hydraulic retention time reached 83.43–97.71% [31]. Therefore, in this study, a wetland planting scheme involving the planting of Water hyacinth, Reed, and Canna in CW was selected to eliminate the effects of β-lactams.

## 3. Materials and Methods

### 3.1. Principles and Sources of Receptor Protein Selection for Characterizing the Combined Biodegradability of β-Lactams

β-lactams show bactericidal properties, as observed in Escherichia coli, by interrupting the formation of bacterial cell walls through covalent binding action with the penicillin-binding protein (PBP) [10]. Combined biodegradation by plants and microbial actions is an important pathway for β-lactam degradation in the environment [10]. Plant degradation of β-lactams is mainly achieved through absorption and mineralization by plants [32,33], whereas the microbial degradation of antibiotic molecules is achieved through enzymatic degradation [2]. The water channel proteins in plant roots mediate the transport of organic molecules in plants, affecting the plant’s ability to absorb β-lactams [34], and the ability of mineralizing enzymes secreted by plants affects the elimination of β-lactams from the aquatic environment [35]. In *Pseudomonas*, plant absorption and mineralization are the main ways of remediating β-lactam contamination by secreted β-lactamase [36]. In this work, four kinds of *E. coli* PBP, plant water channel proteins that affect plant absorption, plant mineralase causing β-lactams mineralization, and *Pseudomonas* β-lactamase related to microbial degradation of β-lactams (6NTZ, 1Z98, 1B85, and 4GZB) were selected as receptor proteins (Figure 5). The receptor protein structures were obtained from the Protein Date Bank (PDB), established by Brookhaven Laboratories, New York, NY, USA.

### 3.2. Combined Biodegradation of β-Lactams Characterization

#### 3.2.1. Characterization of Biodegradability of β-Lactams—Molecular Docking Method

Molecular docking is a common method for analyzing receptor characteristics and the mode of interaction between receptors and ligands, and plays an important role in the study of the binding capacity of small molecular compounds and biomolecules [37]. A larger value of LibDock score indicates a stronger binding interaction between the ligand and the receptor [33]. To characterize the binding effect of β-lactams with receptor proteins, the molecular docking method was applied, where the stereoscopic protein molecules of PBP, plant water channel protein, plant mineralase, and *Pseudomonas* β-lactamase were used as receptors. The LibDock Score of PBP, plant water channel protein, plant mineralase, and *Pseudomonas* β-lactamases were used to characterize the bactericidal properties of β-lactams, plant absorption, mineralization, and microbial degradation, respectively [38].

#### 3.2.2. Characterization of β-Lactams Combined Biodegradation—Comprehensive Matrix Scoring Method with Skewed Weights (CM)

Comprehensive evaluation refers to an overall and holistic evaluation of the object system described by the multi-attribute architecture, where evaluation values are assigned to all the evaluation objects, using a certain method based on the given conditions [39]. The molecular docking of 4 kinds of proteins (6NTZ, 1Z98, 1B85, and 4GZB) was carried out, and the single-factor LibDock Score was obtained. Subsequently, the CM was used to obtain a comprehensive evaluation index (CEI) characterizing the combined biodegradation of β-lactams. The CM was as follows:

(1) Comprehensive evaluation matrix of the combined degradability of β-lactams.

In this work, the affiliation function was selected, and the LibDock Score of β-lactams and four proteins were used as single factor data for comprehensive evaluation matrix construction.
(1)$$R=(r_{ij})=\left[\begin{array}{llll} r_{11} & r_{12} & \cdots & r_{1j}\\ r_{21} & r_{22} & \cdots & r_{2j}\\ \vdots & \vdots & & \vdots \\ r_{i1} & r_{i2} & \cdots & r_{ij} \end{array}\right]$$
where $r_{\mathrm{ij}}=\frac{c_{\mathrm{ij}}-\min\left\{c_{\mathrm{ji}}\right\}}{\underset{j}{\max}\left\{c_{\mathrm{ji}}\right\}\;-\;\underset{j}{\min}\left\{c_{\mathrm{ji}}\right\}}$, which denotes the degree of preference of the *i*th β-lactams regarding the action of the *j*th single factor [40].

(2) Comprehensive evaluation matrix tilted weights of the integrated judgment.

After constructing the comprehensive evaluation matrix for the combined degradability of β-lactams, the weight coefficients A = (a1, a2, a3, a4) were set [41]. In this paper, the weight coefficient tends to be the biodegradation of β-lactamase, that is, the proportion of microbial degradability is 55% and the proportion of other characteristics is 15%, that is, A = (15%, 15%, 15%, 55%).
(2)C=A ∗ R

### 3.3. Molecular Modification of β-Lactams with High Performance Combined Biodegradation Properties Using Quantitative Structure–Activity Relationship (QSAR) Model

In this paper, the quantitative structure–activity relationship (QSAR) was performed using SYBYL-X2.0 software, which was based on the comparative molecular similarity indices analysis (CoMSIA) model for the analysis of the relationship between structure and combined biodegradation of β-lactams [42,43]. Subsequently, a comprehensive model of high-performance combined biodegradation of β-lactams was established. After debugging and screening, the Cefotiam was selected as the template molecule, and its structural part that resembled other compound molecules was chosen as the common backbone for molecular stacking. The 27 molecules were approximately divided into a 3:1 ratio [44,45], with 21 molecules as the training set (containing template molecules) and 7 molecules as the test set (containing template molecules). Partial least squares (PLS) method was used for establishing a constitutive relationship, and for the analysis, the compounds in the training set were cross-validated by the Leave-One-Out method. Then, *q*^2^ and *N* were calculated, followed by regression analysis using No-Validation. Subsequently, R2, standard deviation SEE, and test value *F* were calculated, and the training set that met the model requirements was used for the test set prediction. The *r*^2^_pred_ (>0.6) that met the model requirements was determined, and the construction of the CoMSIA model was completed [37,46]. The CoMSIA model contained an electrostatic field (E), steric field (S), hydrophobic field (H), donor field (D), and an acceptor field (A). According to the principle of contour maps distribution of the CoMSIA model, the combined biodegradation of β-lactams could be improved by: the addition of hydrophilic groups near the white region in the H-field, the addition of hydrophobic groups in the yellow region, the addition of negatively charged groups in the red region of the E-field, the addition of positively charged groups in the blue region, increasing the group in the green region of the S-field, decreasing the group in the yellow region, the addition of hydrogen bond donors in the blue-green region of the D-field, addition of hydrogen bond donors in the purple region, the addition of hydrogen-bonded acceptors in the purplish-red region of the A-field, and the addition of hydrogen-bonded donors in the red region [47]. Penicillin (PNC), which was widely used in breeding because of its better growth-promoting ability in breeding [48], was chosen as the target molecule (Figure 6) for molecular modification (atoms 10, 11, 12, 14, and 15 were the common backbone). The contour maps of the CoMSIA model with PNC as the target molecule is shown in Figure 7.

Considering the information of high performance in combined biodegradation CoMSIA comprehensive model site information and structure-activity relationship information, the 7-, 17-, and 18-site of PNC were selected for molecular modification of β-lactams combined biodegradation. This was followed by the introduction of -N=NH, -NH-NH_2_, -NO, -NO_2_, -N_3_, -ONO, -OCHO, -OOH, -SH, -SO, and -SO_2_ base groups for single, double, and triple site molecular modifications.

### 3.4. Construction of Resistant Mutations of β-Lactam Target Receptor Proteins

#### 3.4.1. Drug Resistant Mutations in *E. coli* PBP Using Sequence Manipulation Suite

It has been shown that resistance in *E. coli* was associated with a conserved sequence in the binding region of the PBP, “SXXK”, which is the key residue and its adjacent amino acids [27]. The sequence manipulation suite was used to randomly mutate the key residue “SXXK” and its adjacent amino acids to affect its drug resistant properties [49] for obtaining the optimal drug resistant amino acid sequence of PBP.

#### 3.4.2. Construction of Drug Resistant Mutations of *E. coli* PBP Using Homology Modeling Algorithms

The homology modeling algorithm is an accepted method for predicting the protein structure of unknown synthetic receptor proteins based on known similar protein structures [50]. With the help of SWISS-MODEL [51,52,53,54,55] in the Automated Protein Modeling Server (APMS) provided by the Glaxo Smith Kline Center in Geneva, Switzerland, the obtained amino acid sequences after drug resistant mutations were submitted to the server, and the calculations were performed by clicking the Build Model. The homology modeling result with the highest sequence identity value (tending to 1) was selected as the alternative protein for the receptor protein resistant mutations. Additionally, the PDB format file of this result was downloaded as the molecular counterpart of the receptor protein for the characterization of antibacterial resistance and resistant mechanism analysis of β-lactam substitute.

### 3.5. System Construction for Integrated Elimination of the Effects of β-Lactams in CW Using Molecular Dynamics Simulation

Eliminating the environmental impact of antibiotics based on CW and combined biodegradation schemes is a better sustainable ecological measure [18]. *Eichhornia crassipes* (Mart.) Solms (Water hyacinth), *Phragmites australis* (Cav.) Trin. ex Steud. (Reed), *Typha orientalis* Presl, and *Canna indica* L. were among the plants that are routinely grown in CW and reduce residual antibiotics in the system [29]. Thus, the wetland plant root exudates were used as the external condition molecules for the subsequent analysis. Since *Pseudomonas*, which is often found in plant rhizosphere, is widely present in the environment [56], *Pseudomonas* β-lactamase was selected as a ligand protein to characterize the microbial degradability of β-lactams in CW. This work constructs a series of CW planting schemes from the perspective of the combined biodegradation of β-lactams to eliminate their environmental impacts.

The current work predicts the degree of influence of molecules from wetland plant root exudates, on the combined biodegradation of β-lactams based on molecular dynamics simulation assisted using molecular docking methods to determine the optimal wetland planting scheme. A Dell Power Edge R7425 server was used to perform molecular dynamics simulations based on the Gromacs 4.6.5 software under the Gromos96 43A1 force field. First, a cube box for placing molecules of β-lactams or their substitute, *Pseudomonas* β-lactamases, and molecules of external conditions of wetland plant root exudates was constructed in the kinetic simulation area, and then the water molecules were added as the environmental medium after the molecules to be calculated were placed into the box. The electroneutral equilibrium of the system inside the box was adjusted by adding Na+ or Cl−. The complex system of β-lactams and substitute with *Pseudomonas* β-lactamase, which was set as a group, was subjected to energy minimization simulations based on the steepest gradient method. The temperature was set at 300 K, and a constant standard atmosphere of 1 bar was set as the pressure bath size, followed by molecular dynamics (MD) equilibrium [57]. In the MD equilibrium phase, the complex was unbound from its position and subjected to the original set temperature and pressure conditions. The frog-jump Newton integration method with 2 fs step length and 100,000 steps as the simulation step number was used to perform kinetic simulation. Finally, the binding energy of the desired calculated system was designed by the Molecular Mechanics/Poisson–Boltzmann Surface Area (MM/PBSA) method [58,59]. With the combined biodegradation of β-lactams as the research objective, a simulated blank control group with no external condition molecules of wetland plant root exudates and an experimental group with a combination of external condition molecules of wetland plant root exudates were used to simulate the binding of β-lactams substitute with *Pseudomonas* β-lactamase. New combinations of external condition molecules of wetland plant root exudates that are favorable to the organism to promote the degradability of β-lactams were screened. When calculating the binding free energy using MM/PBSA, first, the equilibrium trajectories of the protein and antibiotic molecule complexes were sampled, and then, the binding free energies of the complex, protein, and β-lactams were calculated separately [60,61,62].

The binding energy formula is:(3)$$G_{\mathrm{bind}}{=G}_{\mathrm{complex}}{\;-\;G}_{\mathrm{free}-\mathrm{protein}}{\;-\;G}_{\mathrm{free}-\mathrm{ligand}}$$

The binding energy of the molecules in the solution is:(4)$${G=E}_{\mathrm{gas}}{\;-\;\mathrm{TS}}_{\mathrm{gas}}+G_{\mathrm{solvation}}$$

The solvation-free energy can be decomposed into polar and non-polar components, as shown in Equation (4):(5)$$G_{\mathrm{solvation}\;}{=G}_{\mathrm{polar}}+G_{\mathrm{nonpolar}}$$

## 4. Conclusions

In this study, an ecological and sustainable combined control scheme of β-lactams was constructed to effectively reduce the adverse effects on the aquatic environment caused by the irregular discharge of rural small-scale breeding wastewater from source modification and end treatment. The main objectives included: (1) Designing an excellent β-lactam substitute with environmental friendliness, comprehensive degradation properties, and antibacterial resistance based on the 3D-QSAR modeling approach, while integrating the homology modeling algorithm, sequence manipulation suite, and the molecular docking method; (2) Determining the mechanism of resistant mutation of the PBP of *E. coli* and the antibacterial resistant mechanism of β-lactam substitute against the PBP of *E. coli* after resistant mutation; (3) Designing and screening a wetland planting scheme in a CW to further eliminate the effects of β-lactams based on molecular dynamics simulations.

## Figures and Tables

**Figure 1 molecules-27-08434-f001:**
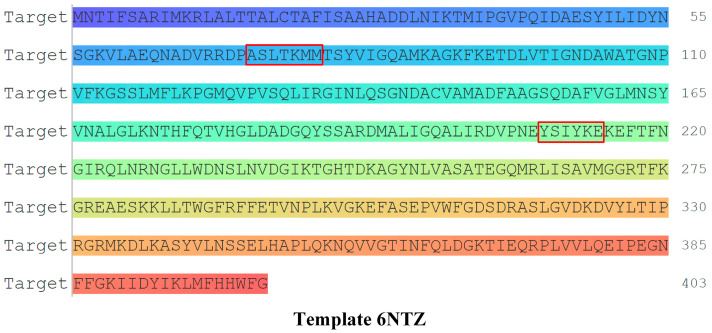
Amino acid sequence of PBP and key residues of drug resistance.

**Figure 2 molecules-27-08434-f002:**
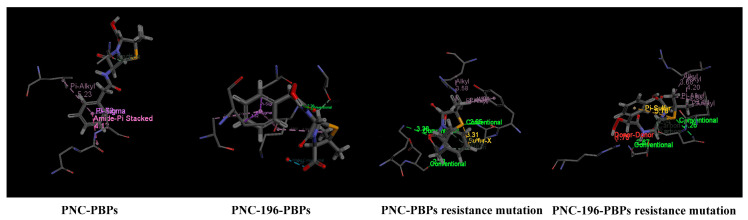
Docking of PNC, PNC-196 to PBP and its resistant mutation.

**Figure 3 molecules-27-08434-f003:**
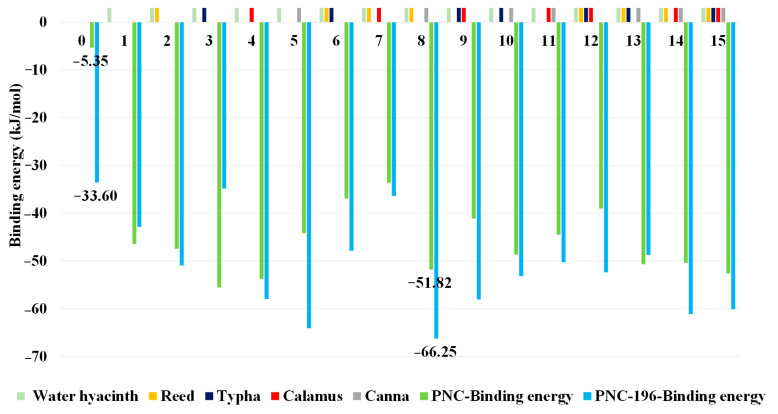
CW to eliminate the effects of β-lactams on wetland plant planting scheme construction and simulation effect characterization.

**Figure 4 molecules-27-08434-f004:**
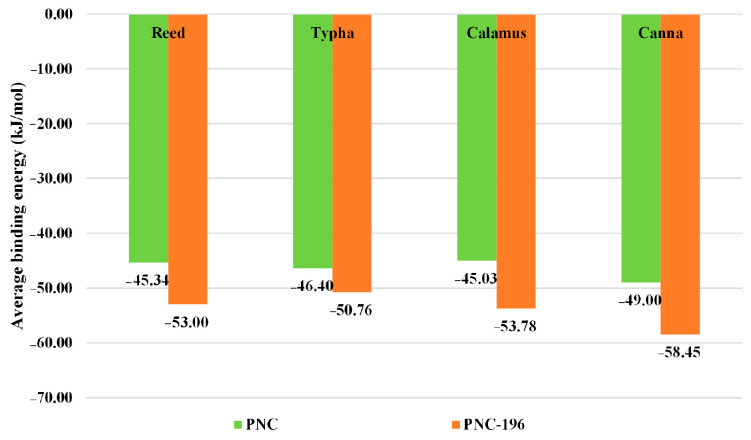
Simulated contribution of wetland plants to the degradation of β-lactams in CW.

**Figure 5 molecules-27-08434-f005:**
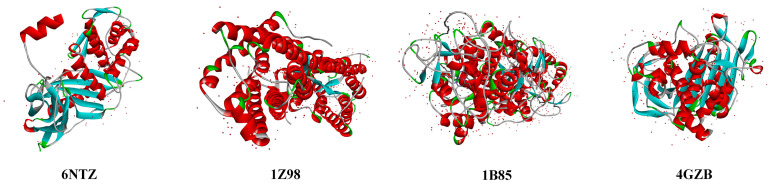
Structure of biodegradable receptor proteins of β-lactams.

**Figure 6 molecules-27-08434-f006:**
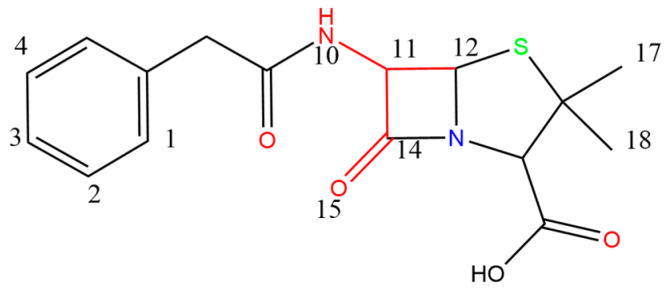
Molecular structure formula of PNC.

**Figure 7 molecules-27-08434-f007:**
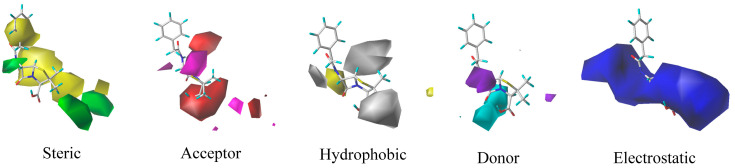
High performance combined biodegradation CoMSIA model contour maps.

**Table 1 molecules-27-08434-t001:** Evaluation of the comprehensive properties of β-lactam substitutes.

No.	Compounds	6NTZ (Å)	Change Rate (%)	1Z98 (Å)	Change Rate (%)	1B85 (Å)	Change Rate (%)	4GZB (Å)	Change Rate (%)
-	PNC	70.00	-	95.11	-	67.49	-	69.72	-
PNC-3	PNC-18-N=NH_2_-17-NH-NH_2_	108.74	55.34	113.58	19.42	78.00	15.57	91.17	30.77
PNC-14	PNC-18-N=NH_2_-7-NH-NH_2_	111.76	15.75	113.56	8.85	82.69	22.51	111.76	15.75
PNC-20	PNC-18-N=NH_2_-7-SO	Failed	-	Failed	-	Failed	-	Failed	-
PNC-21	PNC-18-N=NH_2_-7-SO_2_	Failed	-	Failed	-	Failed	-	Failed	-
PNC-23	PNC-18-NH-NH_2_	105.78	9.57	103.20	−1.08	78.85	16.82	105.78	9.57
PNC-28	PNC-18-NO_2_-17-N=NH_2_	100.72	4.33	106.67	2.25	75.46	11.79	100.72	4.33
PNC-34	PNC-18-NO_2_-17-N=NH_2_-7-OCHO	114.13	18.21	106.30	1.89	84.05	24.53	114.13	18.21
PNC-36	PNC-18-NO_2_-17-N=NH_2_-7-OOH	114.56	18.66	103.20	−1.08	81.57	20.86	114.56	18.66
PNC-43	PNC-18-NO_2_-17-N_3_-7-NH-NH_2_	Failed	-	Failed	-	Failed	-	Failed	-
PNC-47	PNC-18-NO_2_-17-N_3_-7-OOH	Failed	-	Failed	-	Failed	-	Failed	-
PNC-53	PNC-18-NO_2_-17-NH-NH_2_-7-N_3_	Failed		Failed	-	Failed	-	Failed	-
PNC-66	PNC-18-NO_2_-17-NO-7-NO	111.66	15.65	104.66	0.32	89.07	31.97	111.66	15.65
PNC-68	PNC-18-NO_2_-17-NO-7-OCHO	112.12	16.13	107.94	3.46	89.85	33.12	112.12	16.13
PNC-69	PNC-18-NO_2_-17-NO-7-ONO	109.93	13.86	107.25	2.80	88.48	31.10	109.93	13.86
PNC-71	PNC-18-NO_2_-17-NO-7-SH	101.27	4.89	110.64	6.05	83.90	24.31	101.27	4.89
PNC-75	PNC-18-NO_2_-17-NO_2_-7-N=NH_2_	109.58	13.50	110.73	6.14	83.67	23.97	109.58	13.50
PNC-77	PNC-18-NO_2_-17-NO_2_-7-NO_2_	109.72	13.64	111.22	6.61	82.36	22.02	109.72	13.64
PNC-79	PNC-18-NO_2_-17-NO_2_-7-ONO	117.76	21.97	122.57	17.49	86.80	28.60	117.76	21.97
PNC-81	PNC-18-NO_2_-17-NO_2_-7-SH	102.31	5.97	107.86	3.38	79.00	17.05	102.31	5.97
PNC-86	PNC-18-NO_2_-17-ONO	104.62	8.36	112.15	7.50	81.55	20.83	104.62	8.36
PNC-98	PNC-18-ONO-17-NO_2_	107.56	11.40	107.11	2.66	74.38	10.21	107.56	11.40
PNC-108	PNC-18-SO_2_-17-N=NH_2_-7-N_3_	Failed		Failed	-	Failed	-	Failed	-
PNC-118	PNC-18-SO_2_-17-N_3_	Failed		Failed	-	Failed	-	Failed	-
PNC-168	PNC-18-SO_2_-17-SH-7-N_3_	Failed		Failed	-	Failed	-	Failed	-
PNC-196	PNC-7-OOH	78.49	12.13	112.88	18.68	78.22	15.89	92.43	32.06

**Table 2 molecules-27-08434-t002:** Mutations in key residues of PBP resistance and their resistant characterization.

Name	Mutation Zone	Key Residues and Adjacent Amino Acid Sequences	LibDock Score (Å)	Change Rate (%)
PBP	-	ASLTKMM	87.32	-
PBP-1	72–78	HFYFGDA	93.27	−6.82
PBP-2	72–78	QCIDETT	90.76	−3.94
PBP-3	72–78	YIWCQDY	95.91	−9.84
PBP-4	72–78	KPFPVYW	88.03	−0.81
PBP-5	72–78	VRYSVRW	108.51	−24.27
PBP	-	YSIYKE	72.51	-
PBP-6	209–214	VRPDKF	102.90	−41.91
PBP-7	209–214	PWGNSP	108.49	−49.61
PBP-8	209–214	QCSQTT	113.24	−56.16
PBP-9	209–214	MLISTN	104.63	−44.29
PBP-10	209–214	DDPDND	106.83	−47.32
PBP	-	ASLTKMM, YSIYKE	72.51	-
PBP-11	72–78, 209–214	RHRFCHR, WISQRE	114.16	−57.44
PBP-12	72–78, 209–214	REGTLEQ, FEYVRF	Failed	-
PBP-13	72–78, 209–214	WRYSFQW, WESMDS	117.49	−62.02
PBP-14	72–78, 209–214	HVNWSPS, GANEVG	58.11	19.86
PBP-15	72–78, 209–214	CALRDQY, PDGRGW	71.79	1.00
PBP-16	72–78, 209–214	YFHMQNP, CVSPCN	93.22	−28.56
PBP-17	72–78, 209–214	FMYHCQV, FDVLLP	108.13	−49.12
PBP-18	72–78, 209–214	YHEGNAD, KRVRPF	118.59	−63.54
PBP-19	72–78, 209–214	ALVHGNC, HTKRYG	85.59	−18.04
PBP-20	72–78, 209–214	FNELHGF, GVNYCC	Failed	-
PBP-21	72–78, 209–214	NKVFWAD, QWMRPS	Failed	-
PBP-22	72–78, 209–214	YPTSRWV, NRTMVE	Failed	-
PBP-23	72–78, 209–214	KVAEFLT, DDIHKR	92.27	−27.24
PBP-24	72–78, 209–214	EQFHREL, SIARGH	92.07	−26.97
PBP-25	72–78, 209–214	CWCWDLQ, MHAKMI	117.05	−61.41
PBP-26	72–78, 209–214	IHNYKHY, WHYFDI	106.36	−46.68
PBP-27	72–78, 209–214	WIHNIHF, SCDARW	110.36	−52.20
PBP-28	72–78, 209–214	FFTYLPD, HWGTNE	115.45	−59.21
PBP-29	72–78, 209–214	VMEEGNF, GDNDTC	78.39	−8.10
PBP-30	72–78, 209–214	FVSNSSK, AVGQRC	114.91	−58.47
PBP-31	72–78, 209–214	TWCRNST, EIVRNH	Failed	-
PBP-32	72–78, 209–214	ILATADF, NWCFIH	118.67	−63.65
PBP-33	72–78, 209–214	RTNANWG, QALIRD	85.10	−17.35
PBP-34	72–78, 209–214	PKDTEHF, WCKNRT	84.56	−16.61
PBP-35	72–78, 209–214	AVEHVE, EKPHCT	95.27	−31.39

**Table 3 molecules-27-08434-t003:** Evaluation of antibacterial resistance of β-lactam substitutes.

No.	Compounds	LibDock Score (Å)	Change Rate (%)
-	PNC	58.11	-
PNC-3	PNC-18-N=NH_2_-17-NH-NH_2_	63.36	9.04
PNC-14	PNC-18-N=NH_2_-7-NH-NH_2_	70.87	21.96
PNC-28	PNC-18-NO_2_-17-N=NH_2_	Failed	-
PNC-34	PNC-18-NO_2_-17-N=NH_2_-7-OCHO	57.22	−1.53
PNC-66	PNC-18-NO_2_-17-NO-7-NO	31.14	−46.41
PNC-68	PNC-18-NO_2_-17-NO-7-OCHO	67.95	16.93
PNC-69	PNC-18-NO_2_-17-NO-7-ONO	Failed	-
PNC-71	PNC-18-NO_2_-17-NO-7-SH	71.06	22.29
PNC-75	PNC-18-NO_2_-17-NO_2_-7-N=NH_2_	62.13	6.92
PNC-77	PNC-18-NO_2_-17-NO_2_-7-NO_2_	Failed	-
PNC-79	PNC-18-NO_2_-17-NO_2_-7-ONO	Failed	-
PNC-81	PNC-18-NO_2_-17-NO_2_-7-SH	72.62	24.97
PNC-86	PNC-18-NO_2_-17-ONO	Failed	-
PNC-98	PNC-18-ONO-17-NO_2_	43.55	−25.06
PNC-196	PNC-7-OOH	73.04	25.70

**Table 4 molecules-27-08434-t004:** Effects of β-lactams with PBP and its resistant mutation.

**PNC-PBP**			
Hydrophobicity	Hydrophilic		
Non-polar amino acids	Polar amino acids	Polar positively charged amino acids (Basic amino acids)	Polar negatively charged amino acids (Acidic amino acids)
ALA86 ALA151 LEU161	GLN85	-	-
**PNC-196-PBP**			
Hydrophobicity	Hydrophilic		
Non-polar amino acids	Polar amino acids	Polar positively charged amino acids (Basic amino acids)	Polar negatively charged amino acids (Acidic amino acids)
ALA86 ALA89 ALA151 LEU161	-	LYS91	-
**PNC-PBP resistant mutation**			
Hydrophobicity	Hydrophilic		
Non-polar amino acids	Polar amino acids	Polar positively charged amino acids (Basic amino acids)	Polar negatively charged amino acids (Acidic amino acids)
-	TYR54 ASN55	ARG203 LYS323	ASP324
**PNC-196-PBP resistant mutation**			
Hydrophobicity	Hydrophilic		
Non-polar amino acids	Polar amino acids	Polar positively charged amino acids (Basic amino acids)	Polar negatively charged amino acids (Acidic amino acids)
-	TYR54 ASN55	ARG203 ARG373	ASP324

**Table 5 molecules-27-08434-t005:** Key bonds between β-lactams and their substitute with PBP resistant mutation.

**PNC-PBP**				
Bond type	Bonding site	Active bond length	Average bond length	LibDock Score (Å)
Amide-Pi Stacked	GLN85-C1~C6	4.12	3.83	72.51
Pi-Sigma	ALA86-C1~C6	3.72		
Carbon	ALA151-H33	2.25		
Pi-Alkyl	LEU161-C1~C6	5.23		
**PNC-196-PBP**				
Key bonds type	Bonding site	Key bonds length	Average bonds length	LibDock Score (Å)
Pi- Sigma	LEU161-C3~C8	3.02	3.96	76.14
Pi-Alkyl	LYS91-C3~C8	5.11		
Pi-Alkyl	ALA151-C3~C8	4.66		
Alkyl	LYS91-C22	4.99		
Pi-Sigma	ALA86-C3~C8	3.42		
Carbon	ALA89-H35	2.57		
**PNC-PBP resistant mutation**				
Key bonds type	Bonding site	Key bonds length	Average bonds length	LibDock Score (Å)
Conventional	ASP324-H31	2.38	3.38	58.11
Conventional	LYS323-O5	3.38		
Conventional	ASN55-S16	2.95		
Sulfur-X	ASN55-S16	3.31		
Carbon	ASN55-H33	2.95		
Carbon	ASN55-H32	2.96		
Pi-Alkyl	TYR54-H36	4.34		
Pi-Alkyl	TYR54-C20	4.56		
Alkyl	ARG203-C20	3.58		
**PNC-196-PBP resistant mutation**				
Key bonds type	Bonding site	Key bonds length	Average bonds length	LibDock Score (Å)
Pi-Alkyl	TYR54-C21	4.88	3.40	73.04
Pi-Alkyl	TYR54-C22	4.54		
Alkyl	ARG203-C21	3.88		
Alkyl	ARG203-C22	4.20		
Donor-Donor	ARG373-H26	1.75		
Conventional	ASN55-S18	3.26		
Conventional	ASP324-H33	2.67		
Carbon	ASN55-H34	2.82		
Carbon	ASN55-H35	2.64		

## Data Availability

Not applicable.

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
