# Peer review of "Multi-Dimensional Elimination of β-Lactams in the Rural Wetland: Molecule Design and Screening for More Antibacterial and Degradable Substitutes"

_molecules, 2022, doi:10.3390/molecules27238434_

Round 1

Reviewer 1 Report

1. In my opinion, the title is too long, and should be changed.

2. Abstract: Give quantitative details about key results and a brief description about methods used and major conclusion in the abstract.

3. Introduction: The introduction section is repeating well-known and well-established facts. Instead, authors’ focus in this section shall be on importance of this study and how it is going to improve the previous findings. So this section should be modified according to earlier reported studies in β-lactam removal by this technique. Authors should be highlight the novelty of the current study compared to others. This manuscript must be included clear evidence of scientific advancement and novelty in relation to the state of the art. 

4. Tables 1 and 2 are too long, so you can consider putting them in the Support Information.

5. Further, in terms of removal efficiency of β-lactam, is this option really better than hydrolysis, photodegradation or any previous methods known to be applied for similar purposes? Pls make a deep discussion!

Author Response

Response to Reviewers' Comments

(Manuscript Number: molecules-2048025)

The authors would like to thank the editor and reviewers for their careful reading of our manuscript and constructive comments and suggestions. We carefully considered the editor and reviewers’ comments or recommendations and addressed them point-by-point. All the revised sections have been highlighted in YELLOW in the manuscript and the detailed responses to the comments are listed below.

RESPONSES TO THE REVIEWER #1’S COMMENTS

Comment 1: In my opinion, the title is too long, and should be changed.

Response: Thanks for the reviewer’s useful comment. The title has been changed from “Multi-dimensional elimination of β-lactams in the rural wetland: Design of highly degradable substitutions, simulation of strong antibacterial resistance, and enhanced biodegradation of antibiotic substitutions” into “Multi-dimensional elimination of β-lactams in the rural wetland: Molecule design and substitutes screening for more antibacterial and degradable”.

Comment 2: Abstract: Give quantitative details about key results and a brief description about methods used and major conclusion in the abstract.

Response: Thanks for the reviewer’s insightful comment. We have revised the abstract as follow:

Restricted economic and sewage treatment facilities in rural areas lead to the discharge of small-scale breeding wastewater containing higher values of residual beta-lactam antibiotics (β-lactams) which seriously threatens the aquatic environment. In this paper, the molecular docking and a comprehensive method were performed to quantify and fit the source modification for the combined biodegradation of β-lactams. Using penicillin (PNC) as the target molecule, combined with contour maps for substitution modification, a three-dimensional quantitative structure-activity relationship (3D-QSAR) model was constructed for the high-performance combined biodegradation of β-lactams. The selected candidate with better environmental friendliness, functionality, and high performance was screened. By using the homology modeling algorithms, the mutant penicillin-binding proteins (PBPs) of Escherichia coli were constructed to have antibacterial resistance against β-lactams. The molecular docking was applied to obtain the target substitution by analyzing the degree of antibacterial resistance of β-lactams substitution. The combined biodegradation of β-lactams and substitution in the constructed wetland (CW) by different wetland plant root secretions was studied using molecular dynamics simulations. The result showed a 49.28% higher biodegradation of the substitutions than PNC when the combined wetland plant species of Eichhornia crassipes, Phragmites australis, and Canna indica L. were employed.

Comment 3: Introduction: The introduction section is repeating well-known and well-established facts. Instead, authors’ focus in this section shall be on importance of this study and how it is going to improve the previous findings. So this section should be modified according to earlier reported studies in β-lactam removal by this technique. Authors should be highlight the novelty of the current study compared to others. This manuscript must be included clear evidence of scientific advancement and novelty in relation to the state of the art.

Response: Thanks for the reviewer’s valuable recommendation. Part of the introduction section has been rewritten as follow:

For the removal of antibiotics from breeding wastewater, physical adsorption method has the advantages of simple equipment, fast and efficient, convenient operation and management, but activated carbon is difficult to selectively adsorb antibiotics from breeding wastewater with high concentration of organic pollutants [13]; Although the chemical oxidation method has the advantages of high antibiotic clearance rate and strong selectivity in the treatment of antibiotics wastewater, the added chemical oxidants are easy to cause secondary pollution to the aquatic environment [12,14]; Activated sludge technology mainly uses the metabolic action of microorganisms to decompose antibiotics in sewage. Although activated sludge technology has mild reaction conditions and strong operability, the growth environment of microorganisms needs to be strictly controlled, the equipment cost is high, and the treatment range is narrow [11]. The hydrolysis of antibiotics in sewage is not only limited by the temperature and pH of the aquatic environment, but also the influence of ionic strength on the hydrolysis rate of antibiotics is complex and changeable. Different ionic strength may promote or inhibit the hydrolysis of antibiotic molecules, resulting in the hydrolysis reaction of antibiotics in sewage is not easy to be controlled by human [15,16]; The photolysis of antibiotics in sewage is limited by the light absorption capacity of antibiotics and the types of functional groups, etc. For example, penicillin in β-lactams has a weak photolysis reaction in water due to its weak absorption intensity in the near ultraviolet and visible areas [17]. Compared with traditional treatments and natural degradation, constructed wetland has an efficient and stable treatment effect on the removal of antibiotics from breeding wastewater, and on this basis, it has sustainable ecology, better environmental friendliness and lower treatment cost [18].

References:

[11] Alvarino, T.; Suarez, S.; Lema, J.; Omil F. Understanding the sorption and biotransformation of organic micropollutants in innovative biological wastewater treatment technologies. Sci. Total. Environ. 2018, 615, 297-306. DOI: 10.1016/j.scitotenv.2017.09.278.

[12] Bartolomeu, M.; Neves, M.G.P.M.S.; Faustino, M.A.F.; Almeida, A. Wastewater chemical contaminants: remediation by advanced oxidation processes. Photoch. Photobio. Sci. 2018, 17, 1573-1589. DOI: 10.1039/c8pp00249e.

[13] Yu, F.; Han, S.; Ma, J. Adsorptive removal of antibiotics from aqueous solution using carbon materials. Chemosphere 2016, 153, 365-385. DOI: 10.1016/j.chemosphere.2016.03.083.

[14] Rozas, O.; Contreras, D.; Mondaca, M.A.; Pérez-Moya, M.; Mansilla, H. Experimental design of fenton and photo-fenton reactions for the treatment of ampicillin solutions. J. Hazard. Mater. 2010, 177, 1025-1030. DOI: 10.1016/j.jhazmat.2010.01.023.

[15] Mitchell, S.M.; Ullman, J.L.; Teel, A.L.; Watts, R.J. pH and temperature effects on the hydrolysis of three β-lactam antibiotics: ampicillin, cefalotin and cefoxitin. Sci Total Environ. 2014, 466, 547-555. DOI: 10.1016/j.scitotenv.2013.06.027.

[16] Mabey, W.; Mill, T. Critical review of hydrolysis of organic compounds in water under environmental conditions. J. Phys. Chem. Ref. Data 1978, 7, 383-415. DOI: 10.1063/1.555572.

[17] Sheng, F. Transformation and the Associated Toxicity of Penicillin Antibiotics in Natural Water and Soil Mineral Environments. Ph.D. thesis, Nanjing University, Nanjing, P.R. China, 2019.

[18] Hijosa-Valsero, M.; Fink, G.; Schlüsener, M.P.; Sidrach-Cardona, R.; Martín-Villacorta, J. Ternes, T. Removal of Antibiotics from Urban Wastewater by Constructed Wetland Optimization. Chemosphere 2011, 83, 713-719. DOI:10.1016/j.chemosphere.2011.02.004.

Comment 4: Tables 1 and 2 are too long, so you can consider putting them in the Support Information.

Response: Thanks for the reviewer’s comment. We have put Tables 1 and 2 in the Support Information (supplementary file) according to your suggestion.

Comment 5: Further, in terms of removal efficiency of β-lactam, is this option really better than hydrolysis, photodegradation or any previous methods known to be applied for similar purposes? Pls make a deep discussion!

Response: Thanks for the reviewer’s careful review and valuable recommendation. We have added the relevant sections into Introduction.

For the removal of antibiotics from breeding wastewater, physical adsorption method has the advantages of simple equipment, fast and efficient, convenient operation and management, but activated carbon is difficult to selectively adsorb antibiotics from breeding wastewater with high concentration of organic pollutants [13]; Although the chemical oxidation method has the advantages of high antibiotic clearance rate and strong selectivity in the treatment of antibiotics wastewater, the added chemical oxidants are easy to cause secondary pollution to the aquatic environment [12,14]; Activated sludge technology mainly uses the metabolic action of microorganisms to decompose antibiotics in sewage. Although activated sludge technology has mild reaction conditions and strong operability, the growth environment of microorganisms needs to be strictly controlled, the equipment cost is high, and the treatment range is narrow [11]. The hydrolysis of antibiotics in sewage is not only limited by the temperature and pH of the aquatic environment, but also the influence of ionic strength on the hydrolysis rate of antibiotics is complex and changeable. Different ionic strength may promote or inhibit the hydrolysis of antibiotic molecules, resulting in the hydrolysis reaction of antibiotics in sewage is not easy to be controlled by human [15,16]; The photolysis of antibiotics in sewage is limited by the light absorption capacity of antibiotics and the types of functional groups, etc. For example, penicillin in β-lactams has a weak photolysis reaction in water due to its weak absorption intensity in the near ultraviolet and visible areas [17]. Compared with traditional treatments and natural degradation, constructed wetland has an efficient and stable treatment effect on the removal of antibiotics from breeding wastewater, and on this basis, it has sustainable ecology, better environmental friendliness and lower treatment cost [18].

References:

[11] Alvarino, T.; Suarez, S.; Lema, J.; Omil F. Understanding the sorption and biotransformation of organic micropollutants in innovative biological wastewater treatment technologies. Sci. Total. Environ. 2018, 615, 297-306. DOI: 10.1016/j.scitotenv.2017.09.278.

[12] Bartolomeu, M.; Neves, M.G.P.M.S.; Faustino, M.A.F.; Almeida, A. Wastewater chemical contaminants: remediation by advanced oxidation processes. Photoch. Photobio. Sci. 2018, 17, 1573-1589. DOI: 10.1039/c8pp00249e.

[13] Yu, F.; Han, S.; Ma, J. Adsorptive removal of antibiotics from aqueous solution using carbon materials. Chemosphere 2016, 153, 365-385. DOI: 10.1016/j.chemosphere.2016.03.083.

[14] Rozas, O.; Contreras, D.; Mondaca, M.A.; Pérez-Moya, M.; Mansilla, H. Experimental design of fenton and photo-fenton reactions for the treatment of ampicillin solutions. J. Hazard. Mater. 2010, 177, 1025-1030. DOI: 10.1016/j.jhazmat.2010.01.023.

[15] Mitchell, S.M.; Ullman, J.L.; Teel, A.L.; Watts, R.J. pH and temperature effects on the hydrolysis of three β-lactam antibiotics: ampicillin, cefalotin and cefoxitin. Sci Total Environ. 2014, 466, 547-555. DOI: 10.1016/j.scitotenv.2013.06.027.

[16] Mabey, W.; Mill, T. Critical review of hydrolysis of organic compounds in water under environmental conditions. J. Phys. Chem. Ref. Data 1978, 7, 383-415. DOI: 10.1063/1.555572.

[17] Sheng, F. Transformation and the Associated Toxicity of Penicillin Antibiotics in Natural Water and Soil Mineral Environments. Ph.D. thesis, Nanjing University, Nanjing, P.R. China, 2019.

[18] Hijosa-Valsero, M.; Fink, G.; Schlüsener, M.P.; Sidrach-Cardona, R.; Martín-Villacorta, J. Ternes, T. Removal of Antibiotics from Urban Wastewater by Constructed Wetland Optimization. Chemosphere 2011, 83, 713-719. DOI:10.1016/j.chemosphere.2011.02.004.

We have carefully revised all comments in the revised manuscript. Many thanks to the editors and reviewers again.

Best regards,

Yu Li

Reviewer 2 Report

The study is complex and interesting, but I consider that certain terminologies used should be corrected or properly explained.

The abstract must be rewritten. The sentences are too long and in places incomprehensible, too many semicolons are used. The sentences should be shorter and more concise.

Line 34: the abbreviation “β-lactams” already been used in the abstract;

Line 113: Reference 21 is a self-citation that does not seem relevant. The cited article is not related to the mentioned sentence.

Certain tables could be included in the supplementary material, e.g. Table 2.

The quality of figure 5 should be improved.

Lines 367-368: “the resistance of β-lactams increased when the amino acid residues bound to PBP changed from hydrophobic to hydrophilic amino acids”. The amino acid residues bound to PBP or belonging to PBP?

Line 371: “Amino acid residues of β-lactams”? Are you sure this is correct?

The terminology used must be checked. E.g.  (lines-353) “We selected PNC and PNC-196 with significantly enhanced resistance against E. coli”; (line 380) “PNC-196 was 5.01% more resistant to E. coli than the original drug”; (line 381) “the resistance of PNC-196 against E. coli”. The examples mentioned show the resistance of some antibiotics against E. coli. If you can support this with citations, please do so. I only found data from the literature that support the resistance of E. coli to antibiotics, not the other way around. [Hetzer B, Orth-Höller D, Würzner R, Kreidl P, Lackner M, Müller T, Knabl L, Geisler-Moroder DR, Mellmann A, Sesli Ö, Holzknecht J, Noce D, Boonpala O, Akarathum N, Chotinaruemol S, Prelog M, Oberdorfer P. "Enhanced acquisition of antibiotic-resistant intestinal E. coli during the first year of life assessed in a prospective cohort study". Antimicrob Resist Infect Control. 2019 May 20;8:79. doi: 10.1186/s13756-019-0522-6. PMID: 31139362; PMCID: PMC6528363.]

For example, in the following paragraph you present the terms and their meaning correctly. Lines:327-330: “As shown in Table 5, several PBP resistance mutant proteins were constructed using a homology modeling algorithm in this study, and the resistance mutant proteins underwent molecular docking with PNC to characterize the degree of resistance of the mutant proteins. Finally, the most resistant PBP mutants were screened.”

Author Response

Response to Reviewers' Comments

(Manuscript Number: molecules-2048025)

The authors would like to thank the editor and reviewers for their careful reading of our manuscript and constructive comments and suggestions. We carefully considered the editor and reviewers’ comments or recommendations and addressed them point-by-point. All the revised sections have been highlighted in YELLOW in the manuscript and the detailed responses to the comments are listed below.

RESPONSES TO THE REVIEWER #2’S COMMENTS

Reviewer #2: The study is complex and interesting, but I consider that certain terminologies used should be corrected or properly explained.

Response: Thank you very much for your careful review.

Comment 1: The abstract must be rewritten. The sentences are too long and in places incomprehensible, too many semicolons are used. The sentences should be shorter and more concise.

Response: Thanks for the reviewer’s insightful comment. The abstract has been rewritten as follows.

Restricted economic and sewage treatment facilities in rural areas lead to the discharge of small-scale breeding wastewater containing higher values of residual beta-lactam antibiotics (β-lactams) which seriously threatens the aquatic environment. In this paper, the molecular docking and a comprehensive method were performed to quantify and fit the source modification for the combined biodegradation of β-lactams. Using penicillin (PNC) as the target molecule, combined with contour maps for substitution modification, a three-dimensional quantitative structure-activity relationship (3D-QSAR) model was constructed for the high-performance combined biodegradation of β-lactams. The selected candidate with better environmental friendliness, functionality, and high performance was screened. By using the homology modeling algorithms, the mutant penicillin-binding proteins (PBPs) of Escherichia coli were constructed to have antibacterial resistance against β-lactams. The molecular docking was applied to obtain the target substitution by analyzing the degree of antibacterial resistance of β-lactams substitution. The combined biodegradation of β-lactams and substitution in the constructed wetland (CW) by different wetland plant root secretions was studied using molecular dynamics simulations. The result showed a 49.28% higher biodegradation of the substitutions than PNC when the combined wetland plant species of Eichhornia crassipes, Phragmites australis, and Canna indica L. were employed.

Comment 2: Line 34: the abbreviation “β-lactams” already been used in the abstract.

Response: Thanks for the reviewer’s careful review, and following modification has been made in the revised manuscript as follow:

Line: 34: β-lactams, one of the widely used feed additives in the breeding industry, is a commonly used antibiotic [2].

Comment 3: Line 113: Reference 21 is a self-citation that does not seem relevant. The cited article is not related to the mentioned sentence.

Response: Thanks for the reviewer’s comment. Molecular docking method used in our paper was cited from reference 21, as for research content was indeed not related to the mentioned sentence. Thanks for the reviewer’s keen observation.

Comment 4: Certain tables could be included in the supplementary material, e.g. Table 2.

Response: Thanks for the reviewer’s valuable recommendation, we have put Tables 1, 2 and 3 in the supplementary material.

Comment 5: The quality of Figure 5 should be improved.

Response: Thanks for the reviewer’s insightful comment. We have revised Figure 5 as follow:

Figure 5. Docking of PNC, PNC-196 to PBP and its resistant mutation.

Comment 6: Lines 367-368: “the resistance of β-lactams increased when the amino acid residues bound to PBP changed from hydrophobic to hydrophilic amino acids”. The amino acid residues bound to PBP or belonging to PBP?

Response: Thanks for reviewer’s useful comment. We have revised the relevant sections for the questions raised by the reviewer as follow:

Lines: 383-385: When the amino acid residues of binding process of β-lactams to PBP changed from hydrophobic amino acid to hydrophilic amino acid, the resistance of E. coli increased.

Comment 7: Line 367: “Amino acid residues of β-lactams”? Are you sure this is correct?

Response: Thanks for the reviewer’s careful review and valuable recommendation, and the table title “Amino acid residues of β-lactams and its substitution docked to PBP resistance mutation” has been changed into “Effects of β-lactams with PBP and its resistant mutation”.

Comment 8: The terminology used must be checked. E.g. (lines-353) “We selected PNC and PNC-196 with significantly enhanced resistance against E. coli”; (line 380) “PNC-196 was 5.01% more resistant to E. coli than the original drug”; (line 381) “the resistance of PNC-196 against E. coli”. The examples mentioned show the resistance of some antibiotics against E. coli. If you can support this with citations, please do so. I only found data from the literature that support the resistance of E. coli to antibiotics, not the other way around. [Hetzer B, Orth-Höller D, Würzner R, Kreidl P, Lackner M, Müller T, Knabl L, Geisler-Moroder DR, Mellmann A, Sesli Ö, Holzknecht J, Noce D, Boonpala O, Akarathum N, Chotinaruemol S, Prelog M, Oberdorfer P. "Enhanced acquisition of antibiotic-resistant intestinal E. coli during the first year of life assessed in a prospective cohort study". Antimicrob Resist Infect Control. 2019 May 20; 8:79. doi: 10.1186/s13756-019-0522-6. PMID: 31139362; PMCID: PMC6528363.] For example, in the following paragraph you present the terms and their meaning correctly. Lines:327-330: “As shown in Table 5, several PBP resistance mutant proteins were constructed using a homology modeling algorithm in this study, and the resistance mutant proteins underwent molecular docking with PNC to characterize the degree of resistance of the mutant proteins. Finally, the most resistant PBP mutants were screened.”

Response: Thanks for the reviewer’s valuable recommendation. We have revised the relevant sections for the questions raised by the reviewer as follows.

Lines: 369-370: We selected PNC and PNC-196 with significantly enhanced ability against resistant E. coli

 Lines: 395-396: PNC-196 was 5.01% more than the original drug to against resistant E. coli.

Lines: 397: the ability of PNC-196 against resistant E. coli

We have carefully revised all comments in the revised manuscript. Many thanks to the editors and reviewers again.

Best regards,

Yu Li

Reviewer 3 Report

This work proposes a novel idea to eliminate the impact of residual β-lactams of small-scale breeding wastewater on the aquatic environment of rural areas with a sustainable multi-dimensional approach using source modification, antibacterial resistance, and combined wetland plant species-based biodegradation of simulated CW. Overall, this study is an interesting study with comprehensive analysis. Some minor revisions can be made to further improve this manuscript.

1. Why did this study choose 4 proteins including 6NTZ as receptor proteins and complete the construction of comprehensive model based on them? An explanation should be given.

2. In section 2.2.2 of this paper, why does this paper use a mathematical method to comprehensively evaluate the four docking scores obtained to obtain a comprehensive value? What does it mean? An explanation should be given. And. is there any corresponding literature for description?

3. Should a conformation map of protein Ramachandran be given when constructing receptor protein molecules based on homology modeling algorithms?

4. Since Tables 2 and 3 are too large and cumbersome, it is recommended that Tables 2 and 3 be moved to supplementary file.

5. In lines 138-141, how is the weight skewed in the comprehensive method section? And the principle of weight tilt setting shall be explained.

6.  In molecular dynamics simulation, why did the authors choose GROMOS96 43A1 force field? An explanation should be given.

7. In molecular dynamics simulation, how to add background parameters in  simulation area, and what are the characterization substances? An explanation should be given.

Author Response

Response to Reviewers' Comments

(Manuscript Number: molecules-2048025)

The authors would like to thank the editor and reviewers for their careful reading of our manuscript and constructive comments and suggestions. We carefully considered the editor and reviewers’ comments or recommendations and addressed them point-by-point. All the revised sections have been highlighted in YELLOW in the manuscript and the detailed responses to the comments are listed below.

RESPONSES TO THE REVIEWER #3’S COMMENTS

Reviewer #3: This work proposes a novel idea to eliminate the impact of residual β-lactams of small-scale breeding wastewater on the aquatic environment of rural areas with a sustainable multi-dimensional approach using source modification, antibacterial resistance, and combined wetland plant species-based biodegradation of simulated CW. Overall, this study is an interesting study with comprehensive analysis. Some minor revisions can be made to further improve this manuscript.

Response: Thank you very much for your careful review.

Comment 1: Why did this study choose 4 proteins including 6NTZ as receptor proteins and complete the construction of comprehensive model based on them? An explanation should be given.

Response: Thanks for the reviewer’s useful comment. Combined biodegradation by plants and microbial actions is an important pathway for β-lactams degradation in the environment (Chang and Ge, 2021). Plant degradation of β-lactams is mainly achieved through absorption and mineralization by plants (Kumar et al., 2005; Ren et al, 2021), whereas the microbial degradation of antibiotic molecules is achieved through enzymatic degradation (Kumar et al., 2019). The water channel proteins in plant roots mediate the transport of organic molecules in plants, affecting the plant's ability to absorb β-lactams (Berger et al., 2012), and the ability of mineralizing enzymes secreted by plants affects the elimination of β-lactams from the aquatic environment (Gu et al., 2020). In Pseudomonas, plant absorption and mineralization are the main ways of remediating β-lactams contamination by secreted β-lactamase (Zhou, 2019). In this work, four kinds of E. coli PBP, plant water channel proteins that affect plant absorption, plant mineralase causing β-lactams mineralization, and Pseudomonas β-lactamase related to microbial degradation of β-lactams (6NTZ, 1Z98, 1B85, and 4GZB) were selected as receptor proteins.

References:

Kumar, M.; Jaiswal, S.; Sodhi, K.K.; Shree, P.; Singh, D.K.; Agrawal, P.K.; Shukla, P. Antibiotics Bioremediation: Perspectives on Its Ecotoxicity and Resistance. Environ. Int. 2019, 124, 448-461.

Chang, D.H.; Ge, Q.P. Detection and Degradation of Antibiotics in Ecological Environment: a Review. Chinese Agricultural Science Bulletin 2021. 37, 59-64.

Kumar, K.; Gupta, S.C.; Baidoo, S.K.; Chander, Y.; Rosen, C.J. Antibiotic Uptake by Plants from Soil Fertilized with Animal Manure. J. Environ. Qual. 2005, 34, 2082-2085.

Ren, Z.X.; Xu, H.H.; Wang, Y.W.; Li, Y.; Han, S.; Ren, J.B. Combined Toxicity Characteristics and Regulation of Residual Quinolone Antibiotics in Aquatic Environment. Chemosphere 2021, 263, 128301.

Berger, W.A.; Mattina, M.I.; White, J.C. Effect of Hydrogen Peroxide on the Uptake of Chlordane by Cucurbita Pepo. Plant Soil. 2012. 360, 135-144.

Gu, W.W.; Li, X.X.; Li, Q.; Hou, Y.L.; Zheng, M.S.; Li, Y. Combined Remediation of Polychlorinated Naphthalene-Contaminated Soil under Multiple Scenarios: An Integrated Method of Genetic Engineering and Environ-mental Remediation Technology. J. Hazard. Mater. 2020, 405, 124139.

Zhou, P.C. Removal of Antibiotics in Constructed Wetland and its effect on Plant Growth and Sewage Treatment. M.D. thesis, South China Agricultural University, Guangzhou, CHN 2019

Comment 2: In section 2.2.2 of this paper, why does this paper use a mathematical method to comprehensively evaluate the four docking scores obtained to obtain a comprehensive value? What does it mean? An explanation should be given. And is there any corresponding literature for description?

Response: Thanks for the reviewer’s careful review and valuable recommendation. Comprehensive evaluation refers to an overall and holistic evaluation of the object system described by the multi-attribute architecture, where evaluation values are assigned to all the evaluation objects, using a certain method based on the given conditions (Riedel and Pitz, 1986). The molecular docking of 4 kinds of proteins (6NTZ, 1Z98, 1B85, and 4GZB) was carried out, and the single-factor LibDock Score was obtained. The plant absorption, Plant mineralization, microbial degradation and sterilization of β-lactams were characterized by four single-factor LibDock Score, and the combined biodegradability of β-lactams was characterized by fitting these four properties into comprehensive values by the comprehensive matrix scoring method with skewed weights.

References:

Riedel, S.L.; Pitz, G.F. Utilization-Oriented Evaluation of Decision Support Systems. IEEE Trans. Syst. Man Cybern. 1986, 16, 980-996.

Comment 3: Should a conformation map of protein Ramachandran be given when constructing receptor protein molecules based on homology modeling algorithms?

Response: Thanks for the reviewer’s comment. According to the reviewer's suggestion, we have given the protein Ramachandran conformation map, and the Protein Ramachandran maps have put in the supplementary file.

Figure S1. Ramachandran map of PBP resistant mutations.

Comment 4: Since Tables 2 and 3 are too large and cumbersome, it is recommended that Tables 2 and 3 be moved to supplementary file.

Response: Thanks for the reviewer’s valuable recommendation. According to the questions raised by the reviewer, we have put the too long tables in the supplementary file.

Comment 5: In lines 138-141, how is the weight skewed in the comprehensive method section? And the principle of weight tilt setting shall be explained.

Response: Thanks for the reviewer’s insightful comment, and the following changes have made:

Lines: 156-159: After constructing the comprehensive evaluation matrix for the combined degradability of β-lactams, the weight coefficients A = (a1, a2, a3, a4) were set [25]. In this paper, the weight coefficient tends to be the biodegradation of β-lactamase, that is, the proportion of microbial degradability is 55% and the proportion of other characteristics is 15%, that is, A=(15%, 15%, 15%, 55%).

Comment 6: In molecular dynamics simulation, why did the authors choose GROMOS96 43A1 force field? An explanation should be given.

Response: Thanks for the reviewer’s insightful comment. The force field of molecular dynamics (MD) simulation is divided into all-atomic force field, joint atomic force field and coarse particle force field. The quality of the information given by MD simulation is critically depends on the way to describe the interaction between particle atom or atomic group in the system (Martin, 2006). AMBER and CHARMM force field belong to all-atomic force field, which can accurately define the parameters of each atom and are suitable for accurately exploring the forces between ligand and receptor complex systems with high accuracy, but the calculation process is long (Mishra et al., 2021). GROMOS software uses a joint atomic force field, which omits nonpolar hydrogen atoms and incorporating their parameters into adjacent atoms bound to the compounds. GROMOS96 is a good general-purpose force field for proteins, particularly good for free energy perturbations due to soft-core potentials. Weak for reproducing solvation free energies of organic molecules and small peptides. The simulation of GROMOS is efficient and suitable for exploring the interaction between large ligand-receptor protein complex systems (Srivastava et al., 2021). Coarse particle force field further simplifies the position parameters of molecular structure, and the side chain of protein or even the whole amino acid residue is regarded as a particle force field. The simulation results are relatively rough, which cannot meet the purpose of this study (Baran and Mazerski, 2002). In addition, GROMOS96 43A1 was used to simulate the energetics of nucleophile activation in a protein tyrosine phosphatase and etc., which was similar to the complex system of NNIs and resistance/chronic toxicity receptors simulated in this study (Daura et al., 1998; Hansson et al., 1997). Therefore, the GROMOS 9643A1 force field was employed in this paper to simulate the molecular dynamics of NNIs and resistance/chronic toxicity receptors.

References:

Baran, M., Mazerski, J., 2002. Molecular modelling of membrane sterols with the use of the GROMOS 96 force field. Chem. Phys. Lipids. 120, 21-31.

Daura, X., Mark, A.E., Gunsteren, W.F.V., 1998. Parametrization of Aliphatic CHn United Atoms of GROMOS96 Force Field. J. Comput. Chem. 19, 535-547.

Hansson, T., Nordlund, P., Aqvist, J., 1997. Energetics of Nucleophile Activation in a Protein Tyrosine Phosphatase. J. Mol. Biol. 265, 118-127.

Martin, M.G., 2006. Comparison of the AMBER, CHARMM, COMPASS, GROMOS, OPLS, TraPPE and UFF force fields for prediction of vapor–liquid coexistence curves and liquid densities. Fluid Phase Equilibr. 248, 50-55.

Mishra, R.K., Kanhaiya, K., Winetrout, J.J., Flatt, R.J., Heinz, H., 2021. Force field for calcium sulfate minerals to predict structural, hydration, and interfacial properties. Cement Concrete Res. 139, 106262.

Srivastava, I., Kotia, A., Ghosh, S.K., Ali, M.K.A., 2021. Recent advances of molecular dynamics simulations in nanotribology. J. Mol. Liq. 335, 116154.

Comment 7: In molecular dynamics simulation, how to add background parameters in simulation area, and what are the characterization substances? An explanation should be given.

Response: Thanks for the reviewer’s careful review and valuable recommendation. This paper is based on the GROMACS 4.6.5 software in the Dell PowerEdge R7525 server and performs molecular dynamics simulations under the Gromoss9643A1 force field. Firstly, a cube box was built in the dynamic simulation area for placing ligand molecules and target receptors. At the same time, various environmental background factors were added to the box as external conditions, and then water molecules were added as the environmental medium. The simulated system inside the box is electrically neutralized by adding Na+ or Cl-. The energy of the system was minimized by the steepest descent method, and after the energy was converged to 1000kJ/mol, the operations of NVT temperature control (The set temperature is 300K), NPT pressure control (set the pressure to 1 bar to bind the ligand-protein position) and MD balance were performed in sequence. In the MD equilibrium stage, the ligand protein was released from the position constraints under the original set temperature and pressure conditions. The leaping frog Newton integration method was used to simulate the ligand protein with a step size of 2fs. The kinetic simulation with 100000 steps was simulated. Finally, the binding energy of the ligand molecule of the ligand protein monomer and the target receptor was calculated by the Poisson-Boltzmann Surface Are (PBSA) method.

We have carefully revised all comments in the revised manuscript. Many thanks to the editors and reviewers again.

Best regards,

Yu Li